# Fluorescent Magnetic Mesoporous Nanoprobes for Biotechnological Enhancement Procedures in Gene Therapy

**Manuel A. González-Gómez** *,† , **Román Seco-Gudiña** † , **Pelayo García-Acevedo** , **Ángela Arnosa-Prieto**, **Lisandra de Castro-Alves** , **Yolanda Piñeiro** * and **José Rivas**

NANOMAG Laboratory, Applied Physics Department, iMATUS Materials Institute, Universidade de Santiago de Compostela, 15782 Santiago de Compostela, Spain

* Correspondence: manuelantonio.gonzalez@usc.es (M.A.G.-G.); y.pineiro.redondo@usc.es (Y.P.); Tel.: +34-881813097 (M.A.G.G. & Y.P.)

† These authors contributed equally to this work.

**Abstract:** In recent years, nanotechnology has deployed a new set of theragnostic tools, including magnetic resonance contrast agents, nano-delivery systems and magnetic hyperthermia treatments in cancer therapy, exploiting not only the small size of nanoparticles, but also relevant nanoscale properties such as superparamagnetism. Specifically, magnetic nanostructures can be remotely manipulated by external magnetic fields, incrementing their possibilities not only for theragnosis, but also for biotech procedures. Genetic engineering processes involve a set of steps like extracting cells from complex environments, their selection and subsequent cultivation or modification by transfection and can benefit from the use of bioconjugated magnetic nanoparticles. Magnetofection of cells with genes or biological material uploaded on superparamagnetic nanoparticles attracted by a magnetic field greatly increases the efficiency, specificity and speed of the biotechnological procedure in gene transfer systems. This article presents a preliminary investigation into the enhanced transfection efficiency of fluorescent magnetic mesoporous silica nanostructures functionalized with mCherry plasmid, which were used to transfect HeLa cells in just 15 min via magnetic transfection. This method was compared to passive transfection (4 h) and conventional gene transfer using the commercial K2 Transfection System (16 h). The results demonstrated that the fluorescent magnetic mesoporous silica nanostructures were similarly effective to the commercial kit, without the need for reagents that increase costs in clinical therapy. Furthermore, viability assays conducted with HeLa cells showed negligible toxicity at concentrations of up to 50 μg/mL.

**Keywords:** superparamagnetic iron oxide nanoparticles; mesoporous silica nanoparticles; transfection; magnetofection



## 1. Introduction

Gene delivery is an emerging area of biomedicine that involves introducing genetic material into cells for the purpose of gene therapy or research. This field has made significant strides in recent decades towards treating previously untreatable genetic diseases, such as central nervous system disorders and cancer [1–3]. There are several methods for delivering genetic material into cells, including biological strategies and synthetic approaches based on micro and nanotechnology [4–6]. A common biological approach is using inactive viruses as a vehicle to transport genetic material into cells, which can be effective, but can also trigger an immune response or cause unwanted mutations. In addition to the many vector tests required, this method also requires the use of cells as indicators of vector characteristics, which increases variability and requires significant production times, leading to denaturation of genetic materials; therefore, selecting a suitable gene delivery vehicle is critical to the success of cancer gene therapy [7–9]. Synthetic methods, on the other hand, use artificial organic or inorganic nanoparticles as carriers to

transport the genetic material [10]. Organic nanoparticles are made from biocompatible polymers, while inorganic nanoparticles are made from metals (such as gold, silver or iron), metal oxides (such as iron oxides) or more complex materials such as mesoporous silica that have been coated with a biocompatible shell. Although organic nanoparticles have the advantage of being biodegradable, they lack the stability and structural robustness of inorganic nanoparticles, which makes them a better choice for crossing tissue barriers or providing long-term release of genetic material [11].

Among inorganic nanoparticles, superparamagnetic iron oxide nanoparticles (SPIONs) have gained enormous interest for biomedical applications as they can be manipulated remotely using external magnetic fields and can interact in biological processes at nanoscale range. Currently, a variety of SPIONs are in early clinical trials, and some formulations have been clinically approved by the Food and Drug Administration (FDA) for medical imaging, such as magnetic resonance imaging (MRI), magnetic particle imaging (MPI) and therapeutic applications, such as magnetic hyperthermia treatment (MHI) or cell targeting [12–15]. External modification of SPIONs can increase biocompatibility properties, prevent the agglomeration of bare nanoparticles due to electrostatic interactions and provide them with additional surface properties (such as polydopamine, PDA, showing enhanced biocompatibility and optical properties, apart from being used as a surface polymer for pH-dependent release of the anticancer drugs into cells) [8,16–18]. Moreover, SPIONs can be incorporated into other material matrices such as mesoporous silica nanoparticles (MSNs) that have been intensively studied as nanocarriers for prolonged drug delivery due to their highly specific surface area, adjustable particle and mesopore size, good biocompatibility and a versatile silanol surface for the easy grafting of biomolecules. These combined abilities make magnetic mesoporous silica nanostructures robust multimodal platforms that are capable of simultaneously allowing drug and gene delivery [19–21].

The use of complex magnetic nanocarriers offers a great advantage in gene transfection procedures since they can be magnetically forced to effectively and rapidly cross cell barriers compared to passive transfection processes [22]. The application of an external magnetic field on the opposite side of the cell containers exerts a magnetic attraction that promotes a reduction of time and vector dose, allowing a highly efficient transfection with lower cell cytotoxicity [23].

In this work, fluorescent magnetic mesoporous silica nanostructures were developed and tested as efficient magnetic nanocarriers for transfecting HeLa cells through both magnetic and passive uptake. After optimizing transfection conditions, the mCherry plasmid was conjugated to the magnetic nanocarriers to evaluate their potential as a tool for genetic engineering and biotechnology procedures. The efficiency of magnetofection, passive transfection, and conventional gene transfer using a commercial K2® Transfection System (based on liposomes) was studied by transfecting HeLa cells with the nanocarrier plasmid conjugates. The results demonstrated that the fluorescent magnetic mesoporous silica nanostructures were similarly efficient compared to commercially available organic-based kits, such as liposomes, which are used for non-viral nucleic acid delivery into cells. However, the fluorescent magnetic mesoporous silica nanostructures achieved this efficiency in a shorter incubation time and without the need for reagents that increase costs in clinical therapy. In addition, a cell viability assay was used to evaluate the cytotoxicity of the fluorescent magnetic mesoporous silica nanostructures.

## 2. Materials and Methods

### 2.1. Chemicals

All chemicals used in the synthetic processes were used as received without further purification. Iron(III) chloride hexahydrate ($FeCl_3 \cdot 6H_2O$, 99%), iron(II) sulphate heptahydrate ($FeSO_4 \cdot 7H_2O$, 99%), dopamine hydrochloride ($C_8H_{11}NO_2 \cdot HCl$, 98%), ammonium hydroxide ($NH_3$ aq, 25%), N-cetyltrimethylammonium bromide (CTAB, $C_{19}H_{42}BrN$, 98%), sodium hydroxide (NaOH, 98%), tetraethyl orthosilicate (TEOS, $Si(OC_2H_5)_4$, 98%), acetone ($C_3H_6O$, 98%) and isopropyl alcohol (IPA, $C_3H_8O$, 98%) were purchased from MERCK

(Saint Louis, MO, USA). Ethanol ($C_2H_5OH$, absolute grade) was purchased from Panreac (Madrid, Spain). Phosphate-buffered saline (PBS), high-glucose Dulbecco's Modified Eagle's Medium (DMEM) containing GlutaMax, fetal bovine serum (FBS), penicillin-streptomycin and trypsin were obtained from Gibco-Invitrogen (Gibco-Invitrogen[TM]-Fisher, Carlsbad, CA, USA). Cell culture-grade water (Hyclone[TM]-Fisher, Global Ls Soln, Calgary, Canada) was used in all the experiments.

### 2.2. Synthesis of Fluorescent Magnetic MCM-41 Mesoporous Silica Nanostructures (MSNs@SPIONs@PDA)

Multifunctional MSNs@SPIONs@PDA were obtained by subsequential synthetic strategy (co-condensation, co-precipitation and electrostatic interactions processes), described schematically in Figure 1.

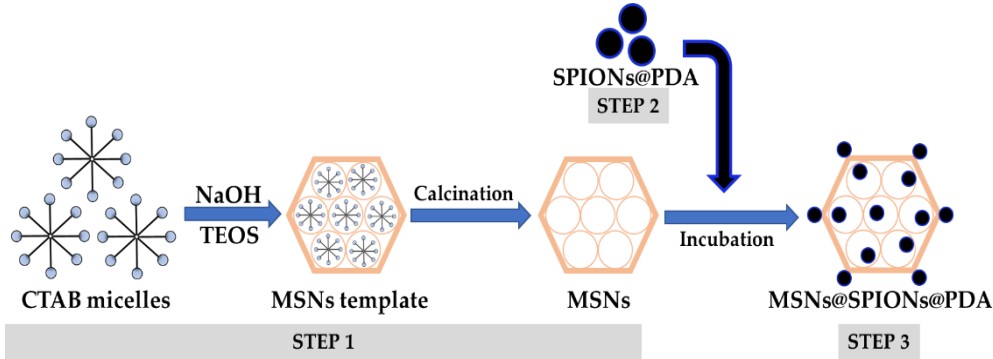

**Figure 1.** Schematic illustration of the synthesized MSNs@SPIONs@PDA obtained through their corresponding step method.

First, MSNs (step 1) were prepared following the co-condensation method described by Gudiña et al. [24], with a slight adjustment of NaOH concentration. In a typical procedure, CTAB (1 g) was dissolved in 480 mL of water under vigorous magnetic stirring (700 rpm). Once homogenized, NaOH (0.98 mL, 6 mM) was added to the CTAB solution, and the reaction was heated at 70 °C for 2 h. Subsequently, TEOS (4.83 mL) was then added dropwise to the mixture, and the reaction remained under magnetic stirring for another 2 h at 70 °C. The thus-obtained mesoporous MCM-41 nanostructures templates were filtered and washed with an ethanol:water 50:50 ($v/v$) mixture (100 mL × 6 times) and dried. The powder was calcined in air at 550 °C for 8 h (heating rate = 1 °C/min) to remove the surfactant from the mesoporous silica matrix.

In parallel, iron oxide nanoparticles coated with a layer of polydopamine, SPIONs@PDA (step 2), were obtained following a modification of a co-precipitation method described by González et al. [25]. In a typical synthesis, $FeCl_3 \cdot 6H_2O$ (12.15 g) and $FeSO_4 \cdot 7H_2O$ (8.35 g) were dissolved in 100 mL of 10 mM HCl aqueous solution with mechanical stirring. The mixture was heated at 60 °C, then $NH_3$ aq (30 mL) and dopamine hydrochloride (2 g) were added, and the reaction was carried out for 1 h. After that, the obtained polydopamine-coated magnetic nanoparticles were acidified up to pH 5 with the incorporation of the HCl solution at 9%. SPIONs@PDA were separated from the reaction medium by a magnetic field and washed several times (6×) with MilliQ water. Finally, SPIONs@PDA were redispersed in Milli-Q water to a final concentration of 5.86% wt (determined by thermogravimetric analyzer, TGA, Figure S1).

MSNs@SPIONs@PDA (step 3) were produced by dispersing 100 mg of MSNs in 5 mL of water. Then, 15 mg of SPIONs@PDA was added and incubated at 70 °C overnight. After the reaction was completed, the fluorescent magnetic material was separated by centrifugation, washed several times (8×) with acetone and water and finally dried at 60 °C for 5 h (134 µg of Fe content, determined by inductively coupled plasma optical emission spectroscopy, ICP-OES).

*2.3. Physicochemical Characterization*

2.3.1. XRD-Structural Characterization

X-ray diffraction (XRD) was performed with a Philips PW1710 diffractometer (PANalytical, Malvern, UK) operated at 40 kV and 30 mA, with a Cu Kα radiation source, λ = 1.54186 Å. Measurements were collected in the 2θ angle range between 10° and 80° with steps of 0.02° and 10 s per step. The mesoporous matrix structure was analysed by low-angle XRD in a PANalytical X'Pert Powder Empyrean (Malvern, UK), in a 2θ range between 0.25 and 6° and a step size of 0.01° (5 s per step).

2.3.2. Surface Chemistry Characterization

Fourier transform infrared (FTIR) spectra of the MSNs@SPIONs@PDA were recorded with a Thermo Nicolet Nexus spectrometer (Thermo Fisher Scientific, Madrid, Spain) using the attenuated total reflectance (ATR) method in the wavenumbers ranging from 4000 to 400 $cm^{-1}$.

2.3.3. Morphological Characterization

The morphology and size of the MSNs@SPIONs@PDA was characterized by transmission electron microscopy (TEM) using a JEOL JEM-1011 microscope operating at 100 kV (JEOL, Tokyo, Japan) and a LIBRA 200FE Transmission electron with High-Resolution Transmission Electron Microscopy operating at 200 Kv (Carl Zeiss Iberia, Madrid, Spain). Samples were placed on copper grids with Formvar® films for analysis. The Image J program (distributed by NIH, Bethesda, MD, USA) was used to measure the diameters of the nanomaterials.

2.3.4. Hydrodynamic Particle Size and Zeta Potential measurements

Measurements of hydrodynamic particle size and ζ-Potential of the MSNs@SPIONs@PDA were routinely performed on the same instrument by a Zetasizer Nano ZS (Malvern Instruments, Malvern, UK) equipped with a He–Ne laser (633 nm) and operating at scattering angle of 173° and at room temperature. All magnetic nanostructures were dispersed in Milli-Q water with magnetic material concentration of 0.4 mg/mL (approx.) and at pH 6.5.

2.3.5. Compositional Characterization

Iron content in the MSNs@SPIONs@PDA samples was determined by ICP-OES (PerkinElmer Optima 3300 DV, Perkin, Waltham, MA, USA).

The composition of the samples was analyzed with a TGA Perkin Elmer model 8000 (Perkin, Waltham, MA, USA).

2.3.6. Magnetic Characterization

Direct current (DC) magnetization curves of dried magnetic samples were measured using a vibrating sample magnetometer, VSM (DMS, Lowell, MA, USA). In such a device, the measurement of magnetic hysteresis loops at room temperature was carried out under external magnetic fields from −10 to 10 kOe.

Temperature dependence measurements of the MSNs@SPIONs@PDA magnetization were made in field-cooled (FC) and zero-field-cooled (ZFC) conditions using a Superconducting Quantum Interference Device (SQUID) Magnetometer (Quantum Design, Darmstadt, Germany).

*2.4. Biological Characterization*

2.4.1. Cells Line

To evaluate the expression of the genetic material, and possible feasibility for the vector, the Human cervical adenocarcinoma cell line–HeLa (ATCC CCL-2™) was used. This cell line was kindly provided by the Galician Oncological Research Group from the Singular Centre for Research in Molecular Medicine and Chronic Diseases (CiMUS, Santiago de Compostela).

Tumor cells were cultured in Dulbecco's modified Eagle medium, DMEM (Gibco Life Technologies, Grand Island, NY, USA) (1:1, *v/v*) and supplemented with fetal bovine serum, FBS, (Gibco®, Life Technologies, Grand Island, NY, USA), 10% and 1% penicillin/streptomycin (Gibco®, Life Technologies, Grand Island, NY, USA) in a humidified atmosphere containing 5% $CO_2$ at 37 °C.

### 2.4.2. Trypsinization

After removing the media from the cells and washing them with a phosphate-buffered saline (PBS) solution (Gibco®, Life Technologies, Grand Island, NY, USA), they were incubated for 10–15 min at 37 °C with a $10\times$ trypsin/EDTA solution (Lonza). Then, the trypsin was neutralized with complete medium. Cells were harvested and centrifuged at 900 rpm for 5 min at room temperature. Finally, the cells were resuspended in complete medium and seeded again. Dilution factors among subcultures were 1:10 for tumor cells, with two passages per week. If it was necessary to calculate the concentration, an aliquot of 10 μL of the total solution (1 mL) was made and they were counted in a Neubauer chamber (Zuzi, Beriáin, Navarra).

### 2.4.3. Freezing

Once centrifuged, the cells were resuspended in the required volume of medium, without supplements, and 10% of DMSO (Sigma-Aldrich, St. Louis, MO, USA) at a final concentration of $1.5 \times 10^6$ cells per millilitre. Cells were stored in a freezer at −80 °C for a period of one week, to later be transferred to a nitrogen tank at a temperature of −187 °C. The passage number of the HeLa cells (ATCC CCL-2™) used in the experiments was 12.

### 2.4.4. Obtaining Genetic Material

Obtained from the Addgene repository from Rob Parton [26] "Addgene plasmid 176016; http://n2t.net/addgene:176016 (accessed on 25 February 2023)" the mCherry plasmid was chosen to undergo genetic procedures.

In order to obtain an optimal concentration of E. coli bacteria, it is necessary to first seed them in a 100 mm plate (Corning®, New York, NY, USA) with a swab through the streaking method. The plate contained agar-agar (MERCK, Saint Louis, MO, USA) enriched for bacteria growth with ampicillin (100 μg/mL). Once seeded, the plate was incubated upside-down for 16 h at 37 °C.

After that time, between three and two colonies were chosen from the plate and grown in 5 mL of lysogeny broth (LB) medium (100 μg/mL, MERCK, Saint Louis, MO, USA), shaking (300 rpm) overnight. Once the medium was sufficiently turbid, a plasmid DNA (5–20 μg) was extracted by performing minipreps.

To carry out the minipreps, the commercial kit EZNA® Plasmid Mini Kit I, Q-spin (Omega, Norcross, GA, USA) was used, following the instructions of the manufacturer.

### 2.4.5. Passive Uptake

Cells were seeded at a density of $1 \times 10^4$ cells/well in a 12-well plate and incubated at 37 °C in 5% $CO_2$ for 24 h. MSNs@SPIONs@PDA were inoculated the following day at 80% confluence, at different concentrations in each well, these being 30, 60 and 100 μg/mL. To avoid the side effects of nanoparticle cytotoxicity, the experiment was performed at all stages with complete DMEM medium. The incubation time lasted 4 h, and after this period the wells were carefully washed with PBS and replaced with fresh complete medium.

### 2.4.6. Magnetic Uptake

Due to the magnetic properties present in the MSNs@SPIONs@PDA, it was proposed to take advantage of their reliable magnetic response by applying an attractive magnetic field to speed up the uptake processes. This would lead to a reduction of incubation times, toxicity and resources.

The principle is based on the magnetic interaction between the magnetic mesoporous nanoprobes, poured over the cells, and a transfection process stimulated by the presence of external permanent magnets located below the culture plate (Figure 2) in order to increase the cellular uptake of the MSNs@SPIONs@PDA, considerably reducing incubation times and, to a certain extent, increasing efficiency.

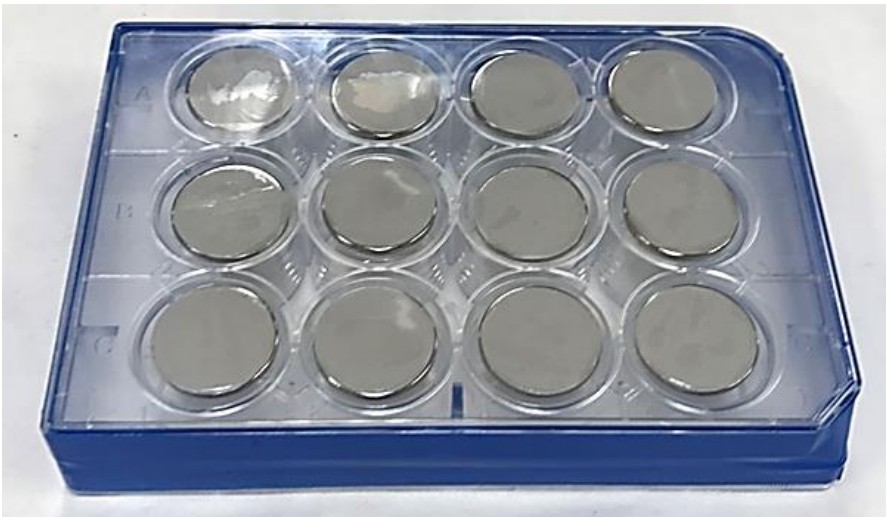

**Figure 2.** Image of custom-made magnetic-based plate used for magnetic uptake adapted to 12-well culture plates.

Identically as in the previous section, the cells were treated in the same way in terms of seeding and preparation. To carry out magnetic uptake, MSNs@SPIONs@PDA were incubated with a fixed concentration (100 μg/mL). Immediately after adding the MSNs@SPIONs@PDA to the medium, a plate with identical dimensions and wells was placed under the plate, which had a neodymium-iron-boron (NdFeB) permanent magnet (strength approx. 3.4 kg, Supermagnete, Gottmadingen, Germany) in each well, coinciding with the upper wells which contained cells and MSNs@SPIONs@PDA. This construction was maintained for 15 min in incubation. At the end of that time, both plates were separated and the wells carefully washed with PBS and filled up again with fresh complete medium.

### 2.4.7. Transfection with Commercial Kit

To evaluate the expression in the cells, transfections were carried out with the plasmid which contained the gene that codes for the red fluorescent protein mCherry. All cells were transfected with the expression vector contained using the K2® reagent (Biontex, München/Laim Germany).

The cells grew in a 12-well plate (Corning®, New York, NY, USA) until reaching 80% confluency. At this time, the transfection was performed. To do this, 3μL of K2® reagent was added in an Eppendorf for each 1 μg of DNA in a final volume of 100 μL of DMEM medium without supplements or antibiotics. The mixture was incubated for 15–20 min at room temperature. After that time, the complete medium present in the plate was removed, the wells were washed with PBS, and DMEM medium without supplements was added, and later the 100 μL of the mixture was added to distribute between two wells so that each one carried 50 μL, and incubation time was carried out for 16 h. After that time, the wells were washed and replaced with fresh complete medium.

Transfection efficiency was analyzed by fluorescence microscopy analysis. All these experiments were performed at an approximate concentration of $2 \times 10^5$ cells per well, at a temperature of 37 °C in 5% $CO_2$ for 24 h before the experiment.

### 2.4.8. Transfection with MSNs@SPIONs@PDA

Cells were grown in a 12-well plate to 80% confluency. At this time, the transfection was performed. To do this, 50 µg/mL of MSNs@SPIONs@PDA was added in an Eppendorf for each 1 µg of DNA in a final volume of 100 µL of DMEM medium without supplements or antibiotics. The mixture was incubated for 30 min at room temperature, with slight agitation, avoiding the precipitation of the particles. After that time, the complete medium present in the plate was removed, the wells were washed with PBS, and DMEM medium without supplements was added, and later the 100 µL of the mixture was added to distribute between two wells so that each one carried 50 µL, and incubation time was carried out for 4 h. After that time, the wells were washed and replaced with fresh complete medium.

### 2.4.9. Magnetofection with MSNs@SPIONs@PDA

Taking advantage of the previously described magnetic properties on the MSNs@SPIONs@PDA, it was decided to carry out a directed transfection by means of attractive magnetic field.

For attractive magnetic field-driven transfection, cells were seeded under the same conditions ($2 \times 10^5$ cells/well in a 12-well plate at 37 °C in 5% $CO_2$ for 24 h). Plasmid DNA was diluted in 50 µL of serum-free medium and supplements, and 50 µg/mL of MSNs@SPIONs@PDA was diluted in another 50 µL of medium without supplements. The dilutions were mixed and incubated for 30 min, with gentle shaking. After this time, the cells were inoculated with the "transfection mix" and incubated for 15 min with the magnetic plate, as described above.

At the end of this time, the samples were washed with PBS and changed with fresh complete medium. The magnetofection procedure used can be better understood with Figure 3.

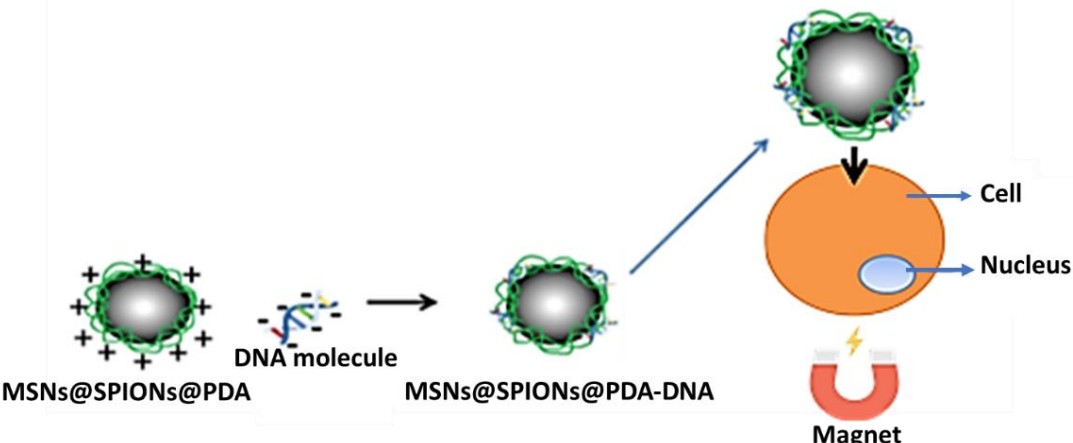

**Figure 3.** Schematic representation of a binding of the genetic material to MSNs@SPIONs@PDA to carry out the magnetofection.

### 2.4.10. Cell Uptake Visualization

All samples were visualized and photographed at a fixed magnification, thus allowing a comparison of transfection efficiency as well as a check for mutations in the cells.

The preparations were first observed under a fluorescence microscope (Zeiss Axio Scope A1, Oberkochen, Germany) and photographed with the Zeiss Axiovision program. For a higher resolution and reliabile visualization, the samples were again observed in a fluorescence confocal multispectral imaging Confocal Laser microscope, Leica TCS SP8 SMD (Leica Microsystems, Kista, Sweden). Technical specifications: Resolution: $1024 \times 1024$; scan direction X: bidirectional; objective: HC PL APO CS 63x/1.40 OIL, using the 405 nm and 552 nm laser lines to PDA and protein chromophore fluorescent visualization, respectively.

### 2.4.11. MTT Assay

Colorimetric MTT [3-(4,5-Dimethylthiazol-2-yl)-2,5-Diphenyltetrazolium Bromide] (Merck, Darmstadt, Germany) assay was used to evaluate the MSNs@SPIONs@PDA cytotoxicity upon exposure to HeLa cell line in 96-well plates, and the medium containing MSNs@SPIONs@PDA at concentrations of 10, 25, 50, 100, 150, 200, 300, 400 µg/mL was added to each well containing $1 \times 10^4$ cells. After 48 h of incubation, the wells were washed with PBS, and a mixture of 4.4 g of the MTT compound diluted in DMEM without supplements was added to a total volume of 8 mL. Each well of the plate would receive 1 mL of the mixture and incubate for 4 h at 37 °C. After that time, the mixture was discarded from the wells gently (avoiding the cells detaching from it) and replaced with 500 µL of DMSO for cell permeation and solubilization of the formazan crystals. To facilitate solubilization, the plate was incubated for 15 min in a dry oven (37 °C).

Absorbance of each well was determined at a wavelength of 570 nm using the Multiskan GO microplate reader (Thermo Scientific™, Waltham, MA, USA). Each experiment was performed in triplicate and in three independent assays. After the assay measurements, *cell viability* values were established in percentages, with data normalized using the following formula [27] (*A* = absorbance):

$$Cell\ viability\ (\%) = \left[ \frac{A_{sample} - A_{blank}}{A_{control} - A_{blank}} \right] \times 100 \tag{1}$$

## 3. Results and Discussion

### 3.1. XRD Patterns

The crystalline properties of the MSNs, SPIONs@PDA and MSNs@SPIONs@PDA were analyzed by recording low and wide-angle X-ray powder diffraction patterns (Figure 4). Low-angle powder XRD patterns from MSNs and MSNs@SPIONs@PDA are shown in Figure 4a. All the diffraction patterns match the three clear Bragg diffraction peaks at (1 1 1), (1 1 0) and (2 0 0), which are characteristic of the ordered mesoporous structure of hexagonal MCM-41 with space group P6 mm [28,29]. Moreover, powder XRD patterns of a wider-angle range of SPIONs@PDA and MSNs@SPIONs@PDA are presented in Figure 4b. All diffraction peaks appeared at different Bragg peaks (2Θ = 30.17°, 35.52°, 43.30°, 53.71°, 57.37°, 62.77°, 71.25° and 74.67°) and corroborated with the Inorganic Crystal Structure Database (ICSD card No. 98-015-8742), which correspond with the (2 2 0), (3 1 1), (4 0 0), (4 2 2), (5 1 1), (4 4 0), (6 2 0) and (5 5 3) planes, respectively, of an inverse spinel structure crystalline phase of magnetite or maghemite [30]. Furthermore, the broad band located between 2Θ = 18° − 25° is related to the short-distance disordered structure of the silica matrix [31].

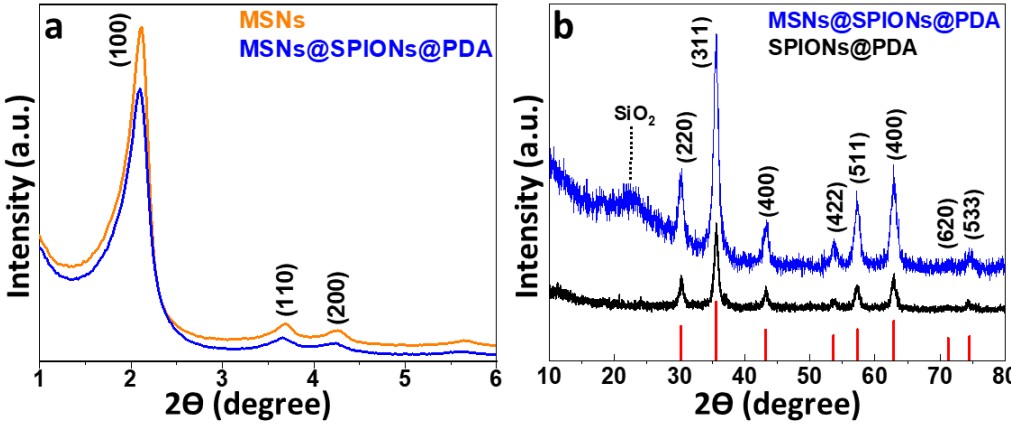

**Figure 4.** (**a**) Low- and (**b**) wide-angle XRD patterns of MSNs (orange pattern), SPIONs@PDA (black pattern) and MSNs@SPIONs@PDA (blue pattern) compared to the XRD pattern of magnetite from the ICSD card No. 98-015-8742 database.

### 3.2. FTIR Spectra

The surface functionalization of the mesoporous silica nanospheres with the fluorescent SPIONS was assessed by FTIR spectra (Figure 5). PDA-coated SPIONs FTIR spectrum confirms the polydopamine coating by the presence of absorption bands at 3366, 2918, 2851, 1611, 1427 and 1051 cm$^{-1}$, associated to -NH stretching, -CH$_2$ (asymmetric and symmetric), -NH bending, aromatic -C=C stretching and -COH symmetric bending vibrations, respectively [32]. On the other hand, the spectrum of MSNs reveals the characteristic bands of a pure-silica material, confirmed by the appearance of three peaks at 1047, 452 and 800 cm$^{-1}$, corresponding to the stretching modes of Si-O-Si (asymmetric and symmetric) and the scissoring vibration of Si-O-Si, respectively [33]. In addition, it can be seen that MSNs@SPIONS@PDA spectrum contains predominantly the siloxane groups and polydopamine absorption bands, as described above. Finally, the peak observed at 525 cm$^{-1}$ is associated with the stretching vibration of the tetrahedral groups for magnetite [34].

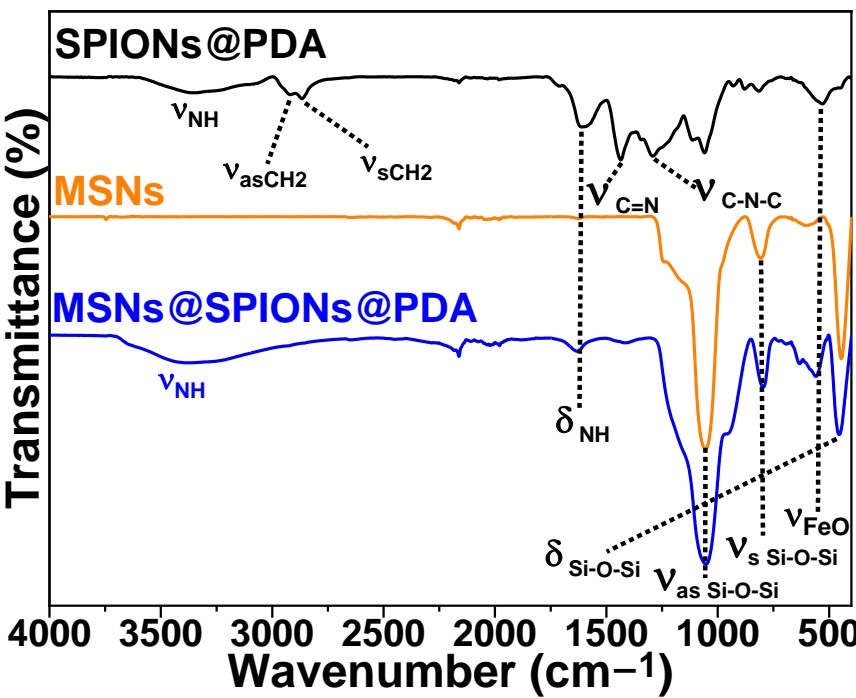

**Figure 5.** FTIR spectra of SPIONS@PDA (black pattern), MSNs (orange pattern) and MSNs@SPIONS@PDA (blue pattern).

### 3.3. TEM and HR-TEM Analysis

The shape and size of the MSN, SPIONS@PDA and MSNs@SPIONS@PDA analyzed by TEM and HR-TEM is shown in Figure 6. All nanostructures appear to be nearly spherical with an average diameter of about 12 nm for the magnetic cores (SPIONS@PDA) and 80 nm for MSNs and MSNs@SPIONS@PDA. Figure 6a contains the TEM image of an individual MCM-41 nanosphere and clearly shows its ordered pore arrangement (pores approximately 3 nm wide) typical of mesoporous silica matrices. On the other hand, PDA-coated SPIONs (Figure 6b) show a small degree of polydispersity with a well-formed crystalline region ornamented with clear regular lattice spacing of 0.20 nm, measured by HR-TEM consistent with the (400) lattice plane of magnetite [35]. Furthermore, as can be appreciated in the TEM image (Figure 6b), the compact structure is formed by the functionalization of the MSNs embedded with SPIONS@PDA.

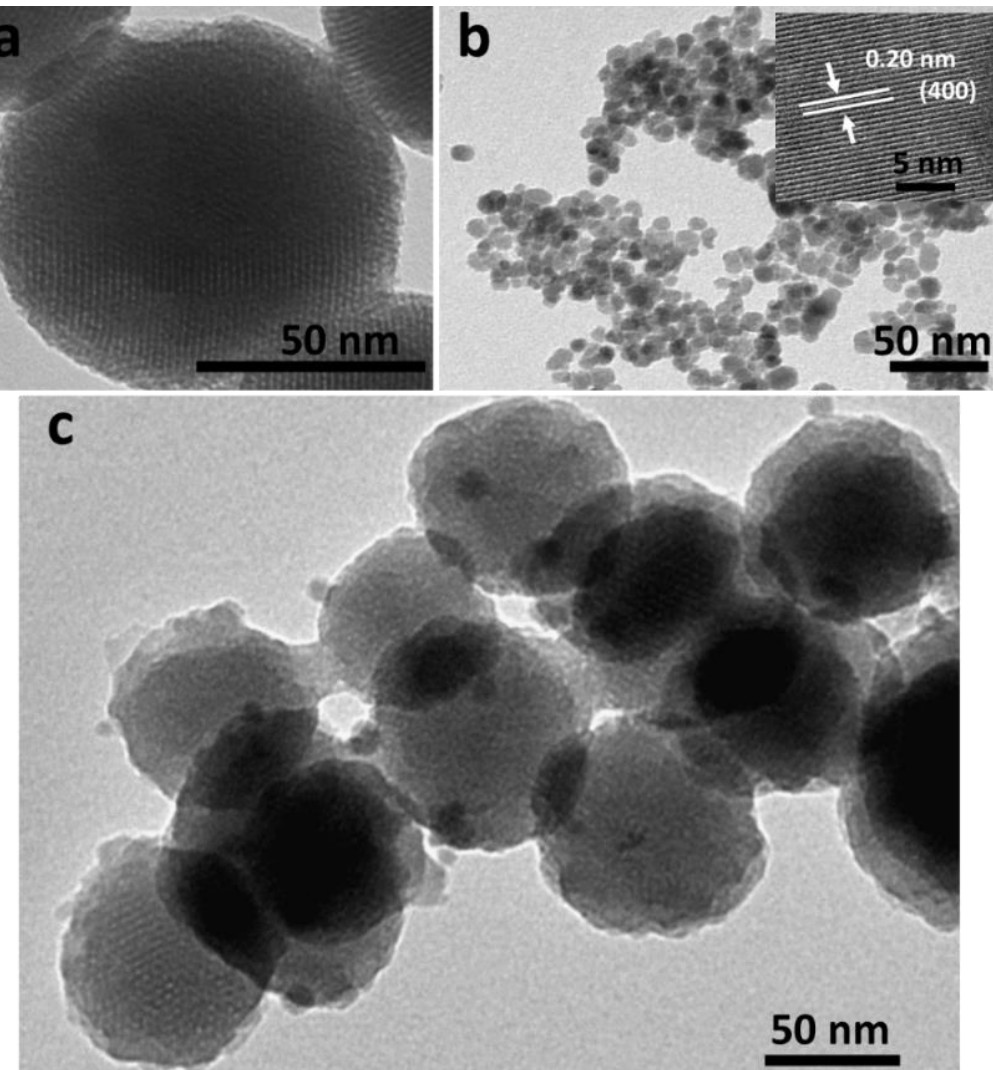

**Figure 6.** TEM micrographs of (**a**) MSNs, (**b**) SPIONs@PDA with HRTEM image of a particle displaying lattice fringes with d-spacings of 0.20 nm, which is characteristic of the (400) of magnetite structure, and (**c**) MSNs@SPIONs@PDA.

*3.4. DLS Characterization*

Colloidal stability of the nanostructures was analyzed by DLS in terms of size, polydispersity index and zeta potential (Table 1; the detailed values for each nanostructure are in Supplementary Figures S2–S4), revealing that both the silica mesoporous nanospheres and the magnetic mesoporous silica nanospheres had a negative zeta potential and a hydrodynamic diameter of about 100 nm and 112 nm, respectively. As expected, DLS size is larger than that measured by TEM due to the scattering of tightly bonded water molecules surrounding the particles [36]. All nanostructures showed PDI values lower than 0.4, indicating their good dispersion in aqueous solution [37]. The zeta potential value of MSNs@SPIONs@PDA (−33 mV), less negative than that found in MSNs (−52 mV), accounts for the positive contribution of amine groups on its surface coming from PDA [38] and further confirms the correct functionalization with SPIONs@PDA (24 mV).

**Table 1.** DLS characterizations of nanostructures: hydrodynamic diameter ($D_H$), polydispersity index (PDI) and zeta potential.

| Sample | $D_H$ (nm) | PDI | Zeta Potential (mV) |
|---|---|---|---|
| **MSNs** | $100.3 \pm 30.5$ | 0.2 | $-52.5$ |
| **SPIONs@PDA** | $77.34 \pm 28.2$ | 0.2 | 24.4 |
| **MSNs@SPIONs@PDA** | $111.7 \pm 47.7$ | 0.3 | $-33.2$ |

### *3.5. Magnetic Properties*

The magnetic behavior of the obtained SPIONs@PDA and MSNs@SPIONs@PDA was analyzed using measurements of magnetic hysteresis loops and zero-field-cooled/field-cooled (ZFC-FC) magnetization curves (Figures 7 and S5). As seen in Figure 7a, the sample displayed negligible coercive fields ($H_C$ = 2.7 Oe) and remanence ($M_R$ = 0.3 emu/$g_{Fe3O4}$), indicating superparamagnetic behavior [39]. Additionally, MSNs@SPIONs@PDA exhibited a good value of saturation magnetization ($M_S$ = 67.1 emu/$g_{Fe3O4}$), which is lower than the bulk magnetite value ($M_S$ = 92 emu/$g_{Fe3O4}$) [34] due to the presence of a dead magnetic layer on small MNPs that reduces the total magnetization [40,41]. The ZFC-FC curves (Figure 7b) also confirmed that the MSNs@SPIONs@PDA are superparamagnetic at room temperature, with a blocking temperature ($T_B$) around 150 K [42].

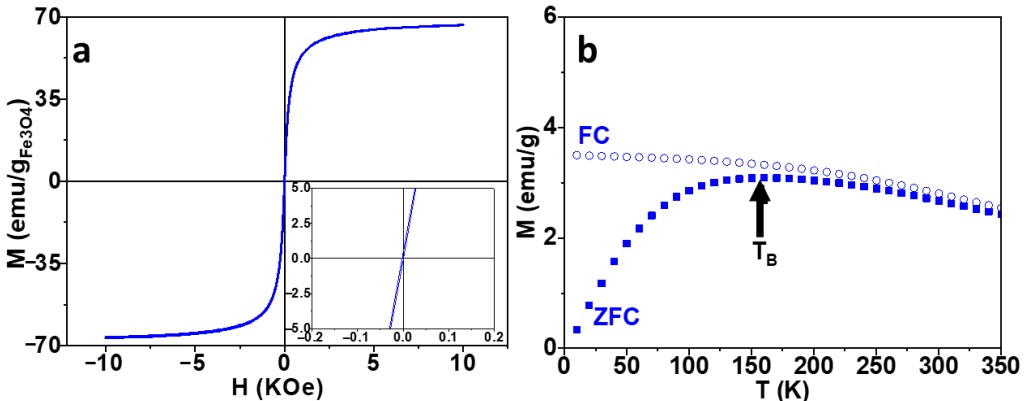

**Figure 7.** (**a**) Magnetic hysteresis loops measured at room temperature by VSM; (**b**) ZFC and FC magnetization curves measured under an applied field of 100 Oe for MSNs@SPIONs@PDA.

### *3.6. Cellular Internalization: Magnetic Uptake vs. Passive Uptake*

The internalization of MSNs@SPIONs@PDA in cells was studied with the help of confocal microscopy (Figure 8). The comparison of bright-field optical images (Figure 8a,b) for both cellular uptake processes shows that magnetic uptake allows more efficient internalization of magnetic mesoporous silica nanospheres than passive uptake, observing better concentration of nanostructures inside the cells. Other notable advantages are related with the cell incubation time period of magnetic (15 min) versus passive (4 h) uptake, since previous studies show that long incubation times of cells, proteins or drugs would lead to their degradation [43–45]. Furthermore, the bright-blue emission of the fluorescent biopolymer PDA can be observed under the excitation of 405 nm laser irradiation (Figure 8c,d), showing a lower internalization efficiency in passive uptake compared to the emission found in the magnetic uptake process, where virtually all cells appear emitting blue.

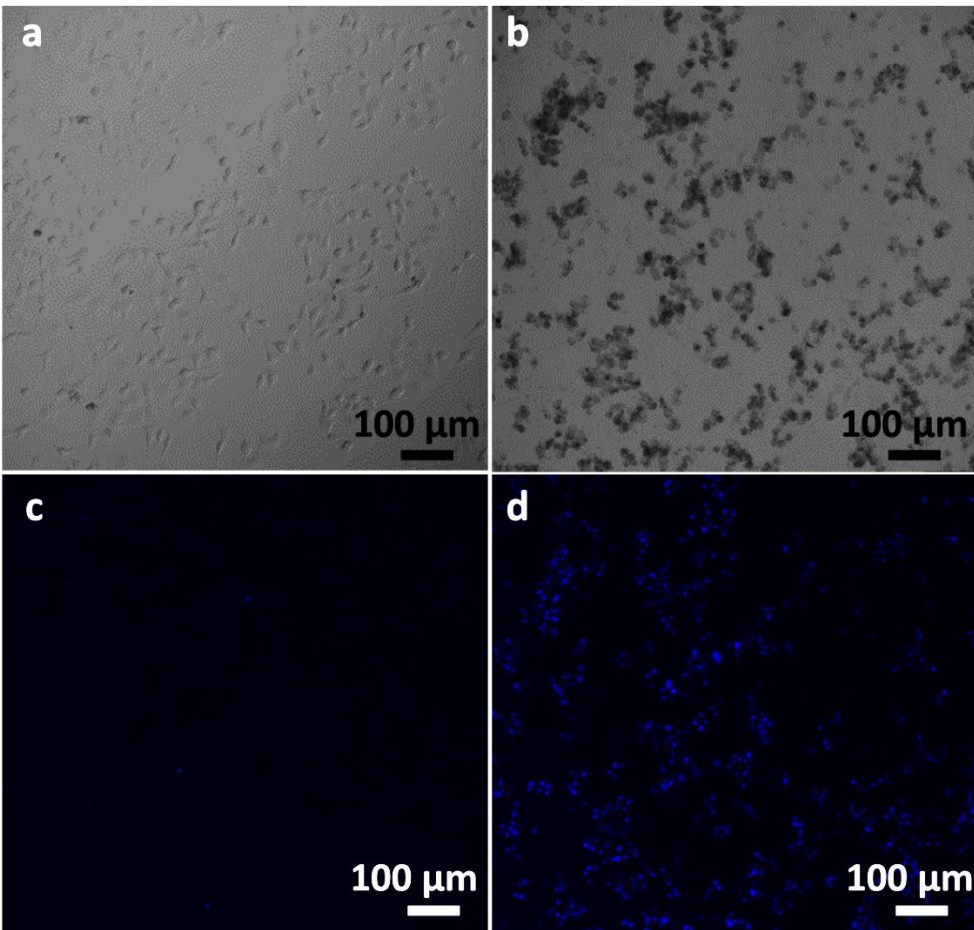

**Figure 8.** Bright-field optical and fluorescent microscopy images of MSNs@SPIONs@PDA internalized in HeLa cells by: (**a**,**c**) passive uptake and (**b**,**d**) magnetic uptake.

### 3.7. *Assessment of Efficacy Transfection: Magnetofection vs. Classical Transfection*

In light of the results obtained in magnetic uptake, another comparative experiment was proposed between a normal transfection using MSNs@SPIONs@PDA, functionalized with plasmid mCherry, and a transfection directed by magnetic force or magnetofection. The results (Figure 9) were compared with a control which carried a commercial transfection agent based on doping cells with cationic liposomes, together with a multiplier, which increased the efficiency of transfection [46,47]. Expression levels can be seen in terms of intensity and efficiency.

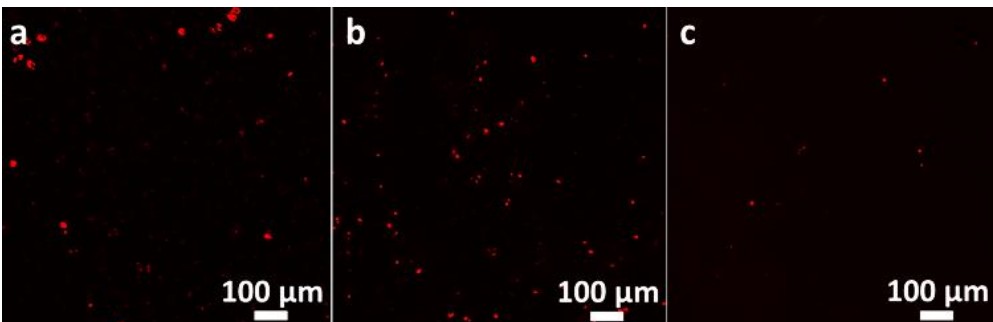

**Figure 9.** Fluorescent detection of plasmid mCherry-transfected HeLa cells by confocal microscopy induced by: (**a**) commercial K2® kit compared with (**b**) cellular magnetofection and (**c**) passive transfection using MSNs@SPIONs@PDA.

For the control group (Figure 9a), the commercial transfection agent was used, and a high amount of transfect cells express a red protein observable in the micrographs. In the second group (Figure 9b), the plasmid was grafted to magnetic nanostructures by simple incubation, and the transfection was carried out by magnetic force (magnetofection). It can be observed that its efficiency is comparable to the control group, but with the advantages of avoiding the use of medium without supplements that might disturb weak cells and of uptake time being highly reduced, which is caused by the good magnetization of MSNs@SPIONs@PDA that experience a strong attraction to external magnets, making cell transfection effective. Finally, a third group of cells transfected with passive uptake of plasmid (Figure 9c) present a noticeable reduction in efficiency with a small amount of red emission spots in the image. Based on the results obtained, magnetofection with complex MSNs@SPIONs@PDA presents not only the clear advantage of short transfection times (see Table 2) providing a better environment for cellular viability without the presence of harmful agents, but also the possibility of delivering internally uploaded cargos. Compared to soft liposomes, complex mesoporous carriers allow for prolonged storage times without degradation, high uploading capacity for long-term delivery of drugs and the grafting of superparamagnetic NPs, as contrast agents in MRI, or magnetic hyperthermia nanoprobes. Stiff and highly porous MSNs@SPIONs@PDA offer advantageous multimodal magnetic abilities that, besides clinical applications [14,20,48], can improve biotech procedures by incrementing efficiency and speed of operating procedures.

**Table 2.** Incubation times comparison of different transfection methods.

| Transfection Method | Incubation Time (min) |
|---|---|
| Passive | 240 |
| Magnetic | 15 |
| Commercial K2® kit | 960 |

### 3.8. Cell Viability

Longitudinal (for 48 h) cell viability assay (Figure 10) allowed a practical approach to compare the biocompatibility of the MSNs@SPIONs@PDA in cell culture media. Determination of viability of HeLa cells after exposition to these nanostructures revealed that up to 50 μg/mL, only a slight reduction in the cell survival (above 80%) was observed in comparison to untreated HeLa cells (0 μg/mL). These results are comparable to or better than other silica mesoporous nanostructures tested in different studies, where the particles were naked or coated with rhodamine, and viability was below 60% even for the lowest concentrations [24,49]. In fact, in recent studies, we have proven that MNPs functionalized with a PDA coating largely increment their biocompatibility as this biopolymer shows low cytotoxicity and low residual biological activity [14]. The additional advantages of being fluorescent, highly desirable for imaging studies and allowing for controlled drug delivery processes [18] makes PDA a smart material that can provide multiple actions for a theragnostic agent with frugal design, promising for efficient targeted gene therapies [50,51].

In fact, PDA coating could be used to enhance biocompatibility of magnetic mesoporous nanostructures, priorly tested as magnetic resonance contrast agents for brain bioim-aging [52] and as the component in controlled drug delivery activity in an hybrid scaffold for bone tissue engineering recovery in vitro [53,54] and in vivo, inserted in a rabbit tibia [55]. As reported in those contexts, magnetic mesoporous nanostructures can be used for the safe transfection of different cell lines, showing a good biocompatibility either with brain microvasculature cells [52], mouse macrophages (cell line RAW 264.7) [53], MC3T3-E1 osteoblast like-cells [54] or even directly inserted as a powder scaffold in a rabbit tibia [55], showing tissue regeneration within 4 months. We have observed that brain endothelial cells [52] cultured for 24 h only show a significant mortality when cultured with large amounts of magnetic mesoporous nanostructures, i.e., 200 μg.

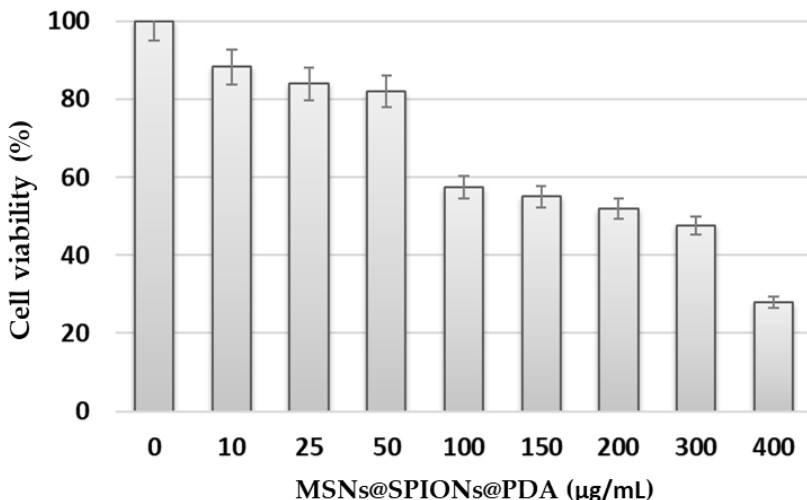

**Figure 10.** Cell viability of HeLa cells measured by the MTT assay (cells were incubated for 48 h with the indicated concentrations of the MSNs@SPIONs@PDA).

## 4. Conclusions

In this work, fluorescent magnetic mesoporous $SiO_2$-MCM-41 nanostructures were synthesized using a three-step synthetic strategy (co-condensation, co-precipitation and electrostatic interactions processes) and physiochemically characterized, showing an average size of 80 nm, good saturation magnetization ($M_S = 67.1$ emu/$g_{Fe3O4}$) and superparamagnetic response adequate for remote magnetic manipulation procedures.

They were incubated with mCherry-modified plasmid, which was attached to the external surface and used as magnetic nanocarriers in different cell uptake procedures to assess the efficiency of magnetic transfection as a valuable biotechnological procedure. Since mCherry-modified plasmid expresses red fluorescence only when they are internalized in the cell nuclei, confocal fluorescence microscopy allows for the precise assessment of the uptake efficiency.

Passive uptake, active uptake (adding a commercial K2® agent) and magnetic uptake (using magnetic dragging with the help of a magnetic cell plate prototype designed ad hoc for the experiments) of the nanocarriers with the plasmid were performed and compared in terms of incubation times and uptake efficiency measured.

It resulted that magnetic uptake allowed for more efficient internalization (in only 15 min) than passive uptake, thus avoiding the long incubation times of the latter that may induce the degradation proteins or other biological materials. Compared to active uptake, magnetic uptake shows similar expression levels of the plasmid but different incubation times (15 min for magnetic uptake and 16 h for active uptake) and is without the requirement of using expensive commercial products.

Therefore, noting that magnetic fluorescent mesoporous silica does not provide cytotoxic effects in the range of concentrations needed for transfection processes, their use for magnetic transfection is promising, taking into account that transfection time is strongly reduced, avoiding the degradation of biological material, and that no expensive additional commercial products are required, which also reduces the expenses of the procedures. Moreover, these nanocarriers offer advantageous multimodal capabilities such as prolonged storage times without degradation, high uploading capacity for long-term delivery of drugs and the ability to provide MRI contrast and magnetic hyperthermia.

**Supplementary Materials:** The following supporting information can be downloaded at: https://www.mdpi.com/article/10.3390/magnetochemistry9030067/s1, Figure S1. TGA thermogram of SPIONs@PDA. A two-stage thermal degradation process for PDA is observed In TGA curve of SPIONs@PDA. First weight loss stage (above 50 °C) is related to the evaporation of water molecules in the biopolymer matrix. The second weight loss occurs around 200 °C, which might correspond

to the thermal oxidation and pyrolysis of the polymer. The residues remaining after pyrolysis are assumed to be the inorganic part of the nanostructures ($Fe_3O_4$ nanoparticles). Figure S2. Dynamic Light Scattering measurements of SPIONs@PDA: Size Distributions by Number (a) and Zeta Potential Distributions (b). Figure S3. Dynamic Light Scattering measurements of MSNs: Size Distributions by Number (a) and Zeta Potential Distributions (b). Figure S4. Dynamic Light Scattering measurements of MSNs@SPIONs@PDA: Size Distributions by Number (a) and Zeta Potential Distributions (b). Figure S5. Hysteresis loops measured on SPIONs@PDA at room temperature.

**Author Contributions:** Conceptualization—M.A.G.-G., R.S.-G. and Y.P.; supervision and formal analysis—Y.P.; writing—M.A.G.-G., R.S.-G., P.G.-A., Á.A.-P., L.d.C.-A., Y.P. and J.R.; investigation and development—M.A.G.-G. and R.S.-G.; supervision and group direction: J.R. All authors have read and agreed to the published version of the manuscript.

**Funding:** This work was supported by the European Commission under the BOW project, (FETPROACT-EIC-05-2019, Grant 952183), CARTsol project, (PLEC2022-009217 funded by MICINN/AEI/under NextGenerationEU/PRTR program) and partially supported by the Spanish Ministry of Science and Innovation (PID2020-112626RB-C21), Modalities «Research Challenges» and «Knowledge Generation» and the Regional Consellería de Innovacion Program for the Grupos de Referencia Competitiva 2021—GRC2021 project of Xunta de Galicia.

**Institutional Review Board Statement:** Not applicable.

**Informed Consent Statement:** Not applicable.

**Data Availability Statement:** Not applicable.

**Acknowledgments:** The authors would like to thank Susana Yáñez Vilar for her help with the magnetic characterizations.

**Conflicts of Interest:** The authors declare no conflict of interest.

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
