# Peer review of "Fluorescent Magnetic Mesoporous Nanoprobes for Biotechnological Enhancement Procedures in Gene Therapy"

_magnetochemistry, doi:10.3390/magnetochemistry9030067_

Round 1
Reviewer 1 Report
The authors presented the paper "Fluorescent magnetic mesoporous nanoprobes for boosting biotechnological procedures in gene therapies"
1) Recent reviews about magnetic nanoparticles and gene delivery should be cited in the work, See the examples fro MDPI system:
https://www.mdpi.com/search?sort=pubdate&page_count=50&q=magnetic++nanoparticles+Gene+delivery&year_from=2022&year_to=2023&featured=&subjects=&journals=&article_types=review-article&countries=
https://www.mdpi.com/2312-7481/9/1/12
https://www.mdpi.com/2079-4991/12/19/3323
https://www.mdpi.com/2624-8549/4/3/63
2) The papers older than 10 years are nor recommended for Introduction section, which shows the perspectives of the area (ref. 4, 6, 12).
3) The novelty of the work should be clearly mentioned in Abstract, Introduction, and Conclusion sections. There are a number of silica-coated magnetic nanoparticles for various applications. The abstract should be focused on the work data instead of various literature data.
4) Section 2..2 should be improved. The mmol of the compounds is very good. However, people usually take mass or volume for the synthesis. For CTAB, TEOS, ferrous salts, etc.
line 124, NH4OH is wrong, better NH3aq (in this solution the major molecule is NH3), concentration of this solution, volume.
minutes = min
Section 2.4.11 MTT
100% cell viability control was buffer instead of nanoparticles? Please, provide information.
5) Table 1. You have so high error for hydrodynamic diameter. Is it a mistake? It is a number mode or intensity. Please, insert the Figures in supplementary or in the text.
It is the error of the average value or peak width?
6) Have you done some calculation of transfection % by flow cytometry?
7) Sections 3.7 please provide the comparison with literature data and your opinion is it good for further applications.
Minor
Section 3.4. Is it possible to provide the magnetization measurement for all synthesized nanoparticles in the work (for fundamental purposes)?
Figure 8, 9 bad pictures contrast and resolution for fluorescent pictures
Author Response
The authors would like to express their gratitude for the valuable review provided by the reviewer, which helped to improve and clarify the manuscript. As a result, we have made the suggested modifications, which have been highlighted in yellow within the text of the manuscript for easy tracking of the changes.

Reviewer 2 Report
Fluorescent magnetic mesoporous nanoprobes were proposed by Manuel A et al. for enhancing biotechnological procedures in gene therapies. In the advanced scientific field, the work design, execution, and explanation of experiments is really basic. However, I appreciate their efforts to envision guided nanomedince techniques in cancer gene therapy and to improve transfection and activity. Before considering publication in the journal Magnetochemistry, some points for clarification are listed below.
1. Change the title of the manuscript to reflect your research hypothesis, is it boosting or enhancing gene transfection compared to commercially available kits? Please use specific words and rewrite the title so that common audiences can easily understand the research.
2. Line 24…This paper demonstrates the effectiveness of fluorescent magnetic mesoporous? If it is increased or decreased, please write it in a manner that is clearly related to your research findings.
3. Line 24..... SiO2-MCM-41 does not need to be mentioned here; simply write Mesoporous silica nanoparticle (MSNs).
Line 28......iron oxide magnetic nanoparticles.....
I couldn't find any words related to iron oxide, so please double-check and explain it clearly, or remove it from the keywords.
4. Please cite the following PDA-related articles and studies.
https://doi.org/10.1021/acsami.1c25180
https://doi.org/10.3390/coatings12010060,
https://doi.org/10.1016/j.jcis.2015.11.001
5. Lines 69–71....The application of an external magnetic field on the opposite side of the cell containers creates a magnetic attraction that promotes a reduction in time and vector dose, allowing for a highly efficient transfection with less cell cytotoxicity...... The reference is missing from this statement. Please include the appropriate citation.
6. Line 102....(methods of co-condensation, co-precipitation, and absorption)???? What exactly is the absorption method? Is it adsorption, absorption, or electrostatic interactions? Please double-check it.
7. Line 172....in the entire manuscript, ICP-MASS and TGA data are not found....why did you include this section in the methods? Please double-check it.
8. Can you please remove lines 217–222 from the manuscript and replace them with the batch number and passage number of the HeLa cells used in your experiments?
9. In your cytotoxicity studies, your nanoparticle (100 ug/mL) shows a 40% reduction in HeLa cell toxicity, but you wrote (line 255) that it reduced toxicity, which contradicts your experiments. Please explain.
10. Because cytotoxicity and cancer studies on a single cell line are inconclusive in the nanomedicine field, please use a different cell line to test the toxicity of your nanocomposites.
11. The explanation of magnetic, passive, and commercial kit cellular uptake data in the discussion section was very poor; please improve it. Furthermore, if you have time, improve the quality of the provided microscopic images.
12. Conclusion -Lines 534-536 need to be revised; please specify which studies were used in your studies; do not add all as if you finished.
Author Response
The authors would like to express their appreciation for the thorough review conducted by the reviewer, which has significantly contributed to the improvement and clarification of the manuscript. All the suggestions provided by the reviewer have been carefully incorporated into the revised version of the manuscript and highlighted in yellow for ease of tracking.

Round 2
Reviewer 1 Report
Thank you for the revised version. Paper may be accepted in present form.